# Cost-effectiveness of diagnostic algorithms including lateral-flow urine lipoarabinomannan for HIV-positive patients with symptoms of tuberculosis

**Nadia Yakhelef** [1]*, **Martine Audibert** [2], **Gabriella Ferlazzo** [3‡], **Joseph Sitienei** [4,5‡], **Steve Wanjala** [6‡], **Francis Varaine** [3‡], **Maryline Bonnet** [1,6‡], **Helena Huerga** [1]

**1** Epicentre, Paris, France, **2** Université Clermont Auvergne, CNRS, CERDI, Clermont-Ferrand, France, **3** Médecins Sans Frontières, Paris, France, **4** Division of National Strategic Health Programs, Ministry of Health, Nairobi, Kenya, **5** Médecins Sans Frontières, Nairobi, Kenya, **6** IRD UMI 233 TransVIHMI—UM–INSERM U1175, Montpellier, France

☯ These authors contributed equally to this work.
‡ These authors also contributed equally to this work.
* nadiayakhelef@hotmail.com

**Data Availability Statement:** All relevant data are within the manuscript and its Supporting Information files.

## Abstract

### Background

Tuberculosis (TB) is the leading cause of death among HIV-positive patients. We assessed the cost-effectiveness of including lateral-flow urine lipoarabinomannan (LF-LAM) in TB diagnostic algorithms for severely ill or immunosuppressed HIV-positive patients with symptoms of TB in Kenya.

### Methods

From a decision-analysis tree, ten diagnostic algorithms were elaborated and compared. All algorithms included clinical exam. The costs of each algorithm were calculated using a 'micro-costing' method. The efficacy was estimated through a prospective study that included severely ill or immunosuppressed (CD4<200cells/μL) HIV-positive adults with symptoms of TB. The cost-effectiveness analysis was performed using the disability-adjusted life year (DALY) averted as effectiveness outcome. A 4% discount rate was applied.

### Results

The algorithm that added LF-LAM alone to the clinical exam lead to the least average cost per TB case detected (€47) and was the most cost-effective with a cost/DALY averted of €4.6. The algorithms including LF-LAM, microscopy and X-ray, and LF-LAM and Xpert in sputum, detected a high number of TB cases with a cost/DALY averted of €6.1 for each of them. In the comparisons of the algorithms two by two, using LF-LAM instead of microscopy (clinic&LAM vs clinicµscopy) and using LF-LAM along with GeneXpert in sputum instead of GeneXpert in urine along with GeneXpert in sputum, (clinic&LAM&Xpert_sputum

**Funding:** The authors received no specific funding for this work.

**Competing interests:** The authors have declared that no competing interests exist.

vs clinic&Xpert_sputum&Xpert_urine) led to the highest increase in the cost-effectiveness ratios (ICERs): €-7.2 and €-12.6 respectively. In these two comparisons, using LF-LAM increased the number of TB patients detected while reducing costs. Adding LF-LAM to smear microscopy alone or to smear microscopy and Xray led to the highest increase in the additional number of TB cases detected (31 and 25 respectively) with an incremental efficiency estimated at 134 and 344 DALYs respectively. The ICERs were €22.0 and €8.6 respectively.

## Conclusion

Including LF-LAM in TB diagnostic algorithms is cost-effective for severely ill or immunosuppressed HIV-positive patients.

## Introduction

Tuberculosis (TB) in one of the leading causes of death worldwide [1]. Of the 10 million estimated incident TB cases in 2017, 9% were among HIV-positive patients [1]. Currently, TB diagnosis relies mainly on microscopic examination of sputum; culture methods on liquid media and molecular methods. Microscopic examination of sputum is a fast, easy and affordable method [2]. However, smear microscopy has low sensitivity (~50%) that is even lower in HIV-positive patients [3–5]. Culture methods on liquid media have a high sensitivity but require a high laboratory infrastructure level, highly qualified staff and scrupulous respect for safety standards. Molecular methods, such as GeneXpert are easy to use, requires little handling and can detect tuberculosis in few hours. It also makes it possible to identify those who have resistance to rifampicin, a very good marker of Multi-drug-Resistant (MDR) TB [6]. WHO recommends this test as an initial test for anyone with signs and symptoms of tuberculosis [7] However, new TB technologies also represent additional cost for national TB programs [8]. In addition, microscopy, culture and GeneXpert, require the production of a sputum sample of sufficient quality and volume to obtain a valid result [9], which presents a challenge in advanced HIV as these patients often have difficulties producing sputum and are more likely to have disseminated forms of TB. In parallel, a new point-of-care urine test, Alere Determine lateral-flow TB lipoarabinomannan Ag (LF-LAM) has emerged as an additional tool with ability to detect TB in advanced HIV-positive patients and presenting advantages over sputum-based testing [9, 10]. LF-LAM is an immunochromatographic test for the qualitative detection of lipoarabinomannan antigen of Mycobacteria. The test is easy to perform and requires limited infection control measures [11]. LF-LAM is performed manually by applying 60μL of urine on the strip and incubating at room temperature for 25 minutes. The intensity of any visible band on the test strip is graded by comparing it with the intensities of the bands on a manufacturer-supplied reference card. LF-LAM utility is greater in HIV-positive patients with advanced immunodeficiency. In a meta-analysis of five studies, the pooled sensitivity and specificity in patients with CD4 below 200 cells/μL was 50% and 90% respectively [12]. WHO currently recommends the test in patients with symptoms of TB and very immunocompromised (CD4<100 cells/μL) or seriously ill [10]. This point-of-care technology can be an efficient tool to aid the diagnosis of TB in resource limited contexts.

Kenya is one of the 30 high TB burden countries with an incidence of 588/100,000 in 2016 [13]. The Nyanza region, where Homa Bay is located, is the area with the highest case load

reported in the country [14]. In Homa Bay, the overall HIV prevalence among people aged 15–49 years was estimated at 20.7% in 2017 [14, 15], more than four times the national average in the country and 74% among TB patients [13–15]. TB and HIV care was provided in Homa Bay District Hospital by the Ministry of Health and *Médecins Sans Frontières* free of charge. HIV testing was proposed to all patients with symptoms of TB. A previous prospective observational study conducted in the same site showed that LF-LAM increased the diagnostic yield in ambulatory and hospitalized, immunocompromised (CD4<200cell/μL) or seriously ill, HIV-positive patients with symptoms of TB [15]. Using data from this cohort study, we aimed to evaluate the cost-effectiveness of introducing urine LF-LAM assay in 10 different TB diagnostic algorithms for this population of HIV-positive patients. We expect that this study would help lower-middle-income countries where all TB technologies are not available to determine how to improve TB diagnosis by looking at the most cost-effective combinations of technologies. The novelty of the study is the comparison of ten diagnostic algorithms combining different diagnostic tools.

## Materials and methods

### Study population

This study uses data from a single- centred, prospective study conducted between October 2013 and August 2015 at the Homa Bay County Hospital (Kenya) [15]. The population of the prospective study consisted on adults (≥15 years) with symptoms of TB hospitalized in the in-patient department or attending the out-patients TB clinic, who were either severely ill, or with a CD4 count below 200cells/μl or a body mass index (BMI) below 17Kg/m², and who had not taken fluoroquinolones or anti-tuberculosis drugs in the month prior to the consultation. On the first day of consultation, the clinical officer performed a clinical exam, a chest X-ray and a LF-LAM test. In addition, two sputum samples (spot and early morning) were collected for smear microscopy, GeneXpert assay and *Mycobacterium tuberculosis complex* (MTBC) culture. For patients not able to produce sputum, GeneXpert and MTBC culture were performed in centrifuged urine from the same sample initially collected for LF-LAM. Smear microscopy and GeneXpert results were given the same day or the day after sample collection. The decision of whether or not to start TB treatment was made by the clinician officer based on the clinical exam, the chest X-ray and the laboratory results. For patients not started on TB treatment a second clinical exam were performed at day 5. The cost-effectiveness analyses included only patients with valid GeneXpert or culture results in sputum or urine. Confirmed TB was defined as positive GeneXpert or MTB culture.

### Ethical approval

The study protocol was approved by the Kenya Medical Research Institute Ethical Review Committee and the Ethical Review Committee at the *Comité de Protection des Personnes* (CPP), Saint-Germain-en-Laye, France15. Before enrolment in the prospective efficacy study, written informed consent was obtained from all adult participants (≥18 years) and from guardians in the case of minors.

### Study model

Using the prospective study efficacy data results of Huerga et al [15], a decision-analysis tree was constructed to determine the cost effectiveness of each TB diagnostic algorithm. In total, 10 algorithms for TB diagnosis were compared (Fig 1). All algorithms contained a clinical exam as the first step. The first group of algorithms included smear microscopy while the

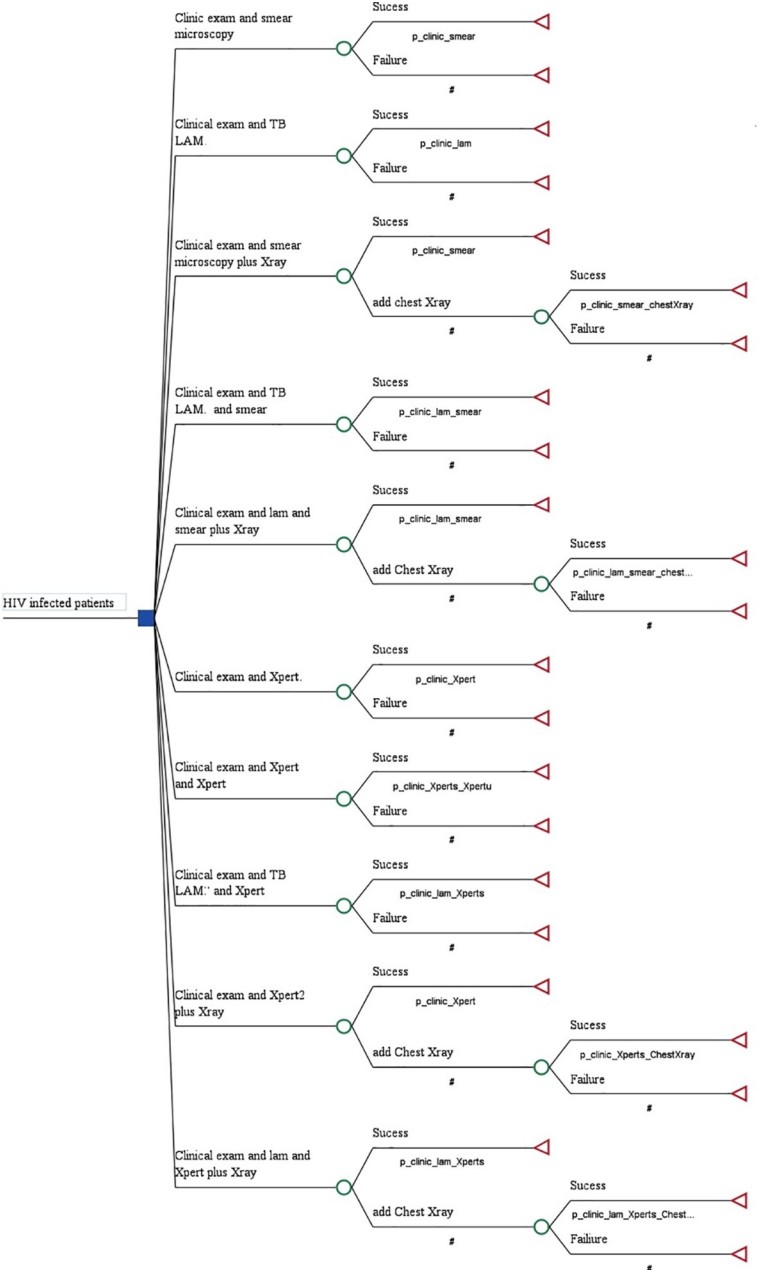

**Fig 1. Diagnostic algorithms using microscopy or GeneXpert alone or in combination with chest X-ray versus LF-LAM alone or in addition to other TB diagnostic tools.**

second group included GeneXpert with and without addition of the urine LF-LAM. Combining clinical exam with either smear-microscopy alone or LF-LAM test alone resulted in algorithms 1 (A1-smear) and 2 (A2- LAM). A1 with additional chest X-ray resulted in algorithm 3 (A3-smear&Xray) and with additional LF-LAM resulted in algorithm 4 (A4-smear&LAM). A1 with both additional chest X-ray and LF-LAM resulted in algorithm 5 (A5-LAM&smear +Xray). Within the second group of algorithms, clinical exam was combined with GeneXpert in sputum, whether alone as algorithm 6 (A6-Xpert) or with additional GeneXpert in urine as in algorithm 7 (A7-Xpert&Xpert_urine). Etiher LF-LAM or chest X-ray or both were added to

GeneXpert in algorithm 8 (A8-LAM&Xpert), in algorithm 9 (A9-Xpert&X-ray) and algorithm 10 (A10-LAM&Xpert&Xray).

All algorithms included a clinic examination. Therefore, to simplify the wording we removed the term "clinic" from the label of the algorithms. The following algorithms were compared between them:

- A2-LAM versus A1-smear

- A4-smear&LAM versus A1-smear

- A4-smear&LAM versus A3-smear&Xray

- A5-LAM&smear&Xray versus A3-smear&Xray

- A4-smear&LAM versus A6-Xpert

- A7-Xpert&Xpert_urine versus A6-Xpert

- A8-LAM&Xpert versus A6-Xpert

- A8-LAM&Xpert versus A7-Xpert&Xpert_urine

- A8-LAM&Xpert versus A9-Xpert&X-ray

- A10-LAM&Xpert+Xray versus A9-Xpert&X-ray

Since LF-LAM, smear microscopy and GeneXpert results were available within 24 hours following the initial consultation, chest X-ray was performed only in cases with a negative result to those.

## Cost estimation

The costing analysis followed the recommendations of International Society for Pharmacoeconomics and Outcomes Research (ISPSOR) [16] and the French Health Authority [17–18]. A health service perspective under real conditions was adopted. The cost evaluation was based on the analysis of the production costs [17]. "Micro-costing" method was adopted for costs of implementing each algorithm [19]. The evaluation of the costs was done by identifying, measuring and valuating the resources used in the production process [17–20]. The same costing approach that Yakhelef et al. [19] was adopted. So, for each test or test prescribed, variable and fixed costs were considered. These costs included both direct (resources used only by the TB diagnostic service) and joint (shared between different services) costs, including variable and fixed costs (depreciation of equipment and buildings). Variable costs estimates were based on expenditures established a posteriori from quantities actually used (consumables, fuel, medicine, actual working time) and from prices in the Kenyan market in 2013 and 2014, using a conversion rate of 113.55 KES for 2013; 118.96 KES for 2014 and 110.12 KES for 2015 (Kenyan shilling) for €1 (www.fxtop.com). Joint costs were calculated based upon allocation keys [17–19]. Costs for clinician, nurse, radiologist and laboratory technician were calculated by multiplying the time spent in the activity by the cost of a unit (minute) of work time or from the average number of patients per day (for supervisory staff). For follow-up of TB treatment, patients had weekly nursing consultations (about 5 min) during the first 2 months, followed by monthly consultations in the last 4 months. Two types of costs were identified. Firstly, the direct variable costs directly attributable to the implementation of each algorithm (consumables, small medical and non-medical equipment). Insofar as certain of these costs were difficult to attribute, an allocation key (proportion of activity for TB diagnosis among total hospital activity) was applied: on the one hand between microscopy (10%), culture laboratory (50%) and for GeneXpert

(40%); one the other hand, between LF-LAM (50%) and CD4 (50%). Secondly, the fixed costs related to the depreciation of medical and nonmedical equipment. For microscopy exam, medical and non-medical equipment were mainly a microscope, a Bio Safety Cabinet, air conditioners and refrigerators. On the other hand, Bio Safety Cabinet Class II was used to prepare the sample for both the GeneXpert test and the MTB culture. Thus, we applied an allocation key according to their respective activity: 40% for GeneXpert test and 60% for the MTB culture. The rest of medical equipment were directly attributable to the GeneXpert (GeneXpert instrument, Battery and Catridges) and culture (Autoclave, Incubator CO2, bath water, etc). For LF-LAM tests, medical material was two Ependorff pipettes. Based on the nomenclature used by the city of Lyon [21], the lifetime for depreciation estimates is 10 years for laboratory equipment, 15 years for air conditioning, 7 years for refrigerators and 25 years for buildings [19]. The total cost was estimated by adding the cost of all categories listed above according to their use in each algorithm. The culture laboratory shared waste water treatment and waste management with the rest of the hospital. For running costs, we used an allocation key related to the surface area. We therefore allocated 4.06% of these total expenditures to the culture laboratory. Afterwards, we allocated 40% of these cost to the GeneXpert (60% for the culture). The costs of the TB treatment (€44.12) for a 6-month rifampicin-based regimen) and Chest X-ray (€1.64/X-ray) were based on a lump sum estimated by MSF. As the cost-effectiveness analysis exceeds one-year time horizon, a 4% discounting rate in order to taking into account the preference for the present was applied, as recommended by the French Health Authority [17].

## Outcomes and measurement of effectiveness

The cost-effectiveness analysis was performed using the disability-adjusted life year (DALY) averted as effectiveness outcome. DALYs are the sum of the Years Lost due to Disability (YLD) and the Years of Life Lost (YLL) due to premature mortality [22]. The YLD in a population are calculated by the number of years persons live with a disability multiplied by a disability weight reflecting the severity of the disability. This weight varies between 0 (no burden) and 1 (mortality). The YLD averted was obtained from the number of TB patients detected by culture or GeneXpert. As recommended by the French Health Authority, the same discount rate of 4% was applied [17]. Parameter estimates for DALY measurement are shown in Table 1.

## Cost-Effectiveness analysis

The diagnostic efficiency of each algorithm was estimated by calculating the incremental cost-effectiveness ratio (ICER) obtained as follows [25]:

$$ICER = \frac{Incremental\ difference\ in\ total\ costs}{Incremental\ difference\ in\ DALYs\ averted}$$

**Table 1. Parameters for DALYs estimation.**

| Parameters | Value | Source |
|---|---|---|
| Age at onset of disability | Individual patient data | [15] |
| Duration of disability *L without treatment (years)* | 1 | [23] |
| Duration of disability *L with treatment (years)* | 11 | [23] |
| Age of death | Individual patient data | [15] |
| Disability weight TB with HIV infection | 0.408 | [23] |
| Life expectancy at age of death | | [24] |
| Discount rate | 0.04 | [17] |

The incremental cost was equal to the difference in terms of cost of implementing an algorithm versus another one. Similarly, the incremental effects were equal to the difference in DALYs averted when implementing an algorithm versus another one [25, 26]. We compared ICERs to a country's willingness-to-pay threshold at €2,673, three times Kenya Gross Annual Income (GNI)[27]. Willingness to pay is the maximum amount a society would be willing to pay to acquire the TB screening technologies considered. The threshold was defined in reference to the country's GNI/capita following standard benchmarks proposed in international work on cost-effectiveness. When ICERs fall below the defined threshold then interventions were considered cost-effective [28–31]. This range of threshold values is generally assumed to encompass the decision makers' willingness-to-pay for an additional unit of effectiveness in health, however much debate still surrounds the determination of an acceptable threshold [32]. To simultaneously account for uncertainty across all parameter inputs, we conducted probabilistic sensitivity analysis using Monte Carlo simulation [33–35]. In each of the 10,000 simulations computed, model inputs were drawn from the data distribution of each parameter (beta probability of effectiveness and Poisson for cost data)[34]. We present the uncertainty around our incremental cost-effectiveness ratios with cost-effectiveness planes [36] and acceptability curves [37] and apply a one-way sensitivity analysis around 0% and 2.5% discount rates [17,18]. Cost-effectiveness calculations and sensitivity analyses were conducted with TreeAge Pro (TreeAge Software, Inc., 2016).

## Results

### Study population

Of the 275 patients included in the cost-effectiveness study, 138 (50.2%) were women, 49 (17.8%) were seriously ill, 149 (54.2%) were on antiretroviral treatment, 127 (46.2%) had a body mass index below 17Kg/m$^2$. Median age was 35 years [IQR: 29–43] and median CD4 count was 113 [IQR: 49–204]. The distribution of the patients according to the level of CD4 count was: 69 (25.6%) <50 cells/μL, 42 (15.6%) 50–99 cells/μL, 83 (30.7%) 100–199 cells/μL, 76 (28.2%) ≥200 cells/μL. In total, 183 (66.6%) patients were hospitalized and 92 (33.5%) were ambulatory. Of the patients with a test result, LF-LAM was positive in 121/275 (44.0%) patients, sputum microscopy in 74/250 (29.6%), Xpert in sputum in 102/251 (40.6%), Xpert in urine in 13/24 (54.2%) and MTB culture in 107/ (44.4%). A total of 156/275 (56.7%) patients had confirmed TB.

### Analysis of costs

The costing details are presented in Table 2. All costs are annualised costs. The main costs were anti-tuberculosis drugs treatment €41.42, followed by the GeneXpert test €8.40, the smear microscopy exam (€3.44); the LF-LAM test (€2.69), the clinical exam (€1.62) and the chest Xray (€1.12). Therefore, the cost of the drug was the main driver of the total cost of TB treatment (91%), the cost of the cartridges was the main component of the cost of GeneXpert test (67%), the cost of human resources was the cost the most important for the clinical and microscopic test (respectively 100% and 70%) and finally, the cost in material and supply represented the most important cost for the LF-LAM test (98%) (Table 2).

### Effectiveness analysis

Of the 275 patients, 97 patients were started on TB treatment through the algorithm A1-smear; 120 through the algorithm A2-LAM; 114 through the A3-smear&Xray; 128 through the A4-smear&LAM; 139 through the A5-LAM&smear&Xray; 116 through the A6-Xpert; 127

**Table 2. Costing details, in Euros (4% discount).**

| Parameters items | Unit cost (€) | % |
|---|---|---|
| **Clinical exam cost** | | |
| Human resources | 1.62 | 100 |
| **Sputum exam** | | |
| Human costs | 2.43 | 70.64 |
| Laboratory maintenance and running | 0.30 | 8.72 |
| Material and furniture | 0.33 | 9.59 |
| Laboratory equipment | 0.21 | 6.10 |
| Non laboratory equipment | 0.17 | 4.94 |
| **Total** | **3.44** | **100** |
| **LF-LAM** | | |
| Human costs | 0.03 | 1.24 |
| Material and furniture | 2.63 | 97.65 |
| Laboratory equipment | 0.03 | 1.11 |
| **Total** | **2.69** | **100** |
| **Chest X-ray** | | |
| Human costs | 0.14 | 12.5 |
| Lump sum Xray | 0.98 | 87.5 |
| **Total** | **1.12** | **100** |
| **GeneXpert** | | |
| Human cost | 0.78 | 9.32 |
| Training | 0.05 | 0.60 |
| Material and furniture | 0.94 | 11.19 |
| Cartridges | 5.64 | 67.12 |
| Running cost | 0.62 | 7.38 |
| Laboratory equipment | 0.21 | 2.50 |
| Non laboratory equipment | 0.03 | 0.36 |
| Infrastructure | 0.13 | 1.55 |
| **Total** | **8.40** | **8.40** |
| **Tuberculosis treatment** | | |
| Human resources costs | 3.33 | 8.04 |
| Drugs | 37.65 | 90.88 |
| Laboratory maintenance and running | 0.15 | 0.36 |
| Material and furniture | 0.21 | 0.51 |
| Laboratory equipment | 0.06 | 0.14 |
| Non laboratory equipment | 0.03 | 0.07 |
| **Total** | **41.43** | **100** |

through the A7-Xpert&Xpert_urine; 137 through the A8-LAM&Xpert versus; 130 through the A9-Xpert&X-ray; 147 through the A10-LAM&Xpert&Xray (Table 3).

## Cost-effectiveness analysis

Table 3 presents the cost and DALYs/algorithm. The mean annualised cost of care for the 275 patients ranged from €5,612 (bootstrap 95% CI €3,495; 8,083) for the A2-LAM algorithm to €9,758 for the A10-LAM&Xpert&Xray (bootstrap 95% CI €6,940; 12,901). The mean DALYs ranged from 1,118 (bootstrap 95% CI €991; 1,220) from the A1-smear algorithm to 1,651 for the A10-LAM&Xpert&Xray (bootstrap 95% CI €1,388; 1,797). The two algorithms leading to the highest average cost/TB case detected were the algorithms that included more than 2

**Table 3. Total cost and total DALYs averted (in euros, 4% discount, N = 275).**

| Algorithm (All algorithms include clinical exam as the first step in the diagnostic procedure) | Cost/TB case detected | Total TB detected confirmed by culture or GeneXpert | Costs (C) (sd) 95%† | DALYS (E) (sd) 95%† | Cost/ Dalys |
|---|---|---|---|---|---|
| A1- Smear | 58.75 | 97 | 5698.79 (1168.22) [3578.91; 8075.35] | 1117.52 (60.52) [990.54; 1219.66] | 5.10 |
| A2- LAM | 46.76 | 120 | 5612.27 (1170.96) [349.53; 8063.86] | 1220.34 (154.41) [895.82; 1487.38] | 4.60 |
| A3- Smear&Xray | 61.27 | 114 | 6985.27 (1276.50) [4633.90; 9622.90] | 1241.79 (42.20) [1139.18; 1308.20] | 5.63 |
| A4-Smear&LAM | 61.85 | 128 | 7917.63 (1467.62) [5235.30; 10962.78] | 1252.03 (180.68) [868.91; 1561.51] | 6.32 |
| A5- LAM&smear&Xray | 69.91 | 139 | 9718.50 (1542.25) [6862.41; 12896.07] | 1586.13 (93.59) [1355.74; 1711.43] | 6.13 |
| A6- Xpert_sputum | 48.39 | 116 | 5613.61 (1158.80) [3482.65; 8035.22] | 1193.12 (131.33) [914.21 ; 1415.83] | 4.70 |
| A7- Xpert_sputum&Xpert_urine | 63.57 | 127 | 8073.62 (1793.56) [4838.89 ; 11817.48] | 1217.66 (136.58) [929.50 ; 1451.09] | 6.63 |
| A8-LAM&Xpert_sputum | 57.28 | 137 | 7848.52 (1482.61) [5170.56; 10933.14] | 1284.57 (209.30) [843.59; 1639.10] | 6.11 |
| A9- Xpert_sputum&X-ray | 54.31 | 130 | 7061.09 (1274.41) [4687.28; 9691.11] | 1431.73 (73.14) [1247.47 ; 1529.79] | 4.93 |
| A10- LAM&Xpert_sputum&Xray | 66.38 | 147 | 9758.26 (1541.70) [6885.31; 12931.95] | 1651.43 (107.11) [1388.34; 1796.55] | 5.91 |

†CIs estimated using 10,000 non-parametric bootstrapping replicates

diagnostic tools in addition to the clinical exam: A10-LAM&Xpert_sputum&Xray (€66) and A5-LAM&smear&Xray algorithm (€70). These were also the algorithms that diagnosed the highest number of TB cases. The two algorithms leading to the least average cost/TB case detected were: A2-LAM (€47) and A6-Xpert_sputum (€48). These two algorithms detected a

**Table 4. Incremental cost, incremental DALYs averted and ICER.**

| Algorithm (All algorithms include clinical exam as the first step in the diagnostic procedure) | Proportion of detected patients | ΔC (sd) (€, 95% CI†) | ΔE (Dalys) (sd) (95% CI†) | ICER (sd) (95% CI†) |
|---|---|---|---|---|
| A2- LAM vs. A1- Smear | +23 | -86.52 (1188.88) [-2412.32; 2274.95] | 102.82 (165.34) (-238.61; 395.47) | - 7.19 (630.68) [-61.57 ; 62.93] |
| A4-Smear&LAM vs A1- Smear | +31 | 2218.84 (1012.27) [-292.36; 4277.66] | 134.51 (189.68) (-263.24; 470.41) | 21.96 (1109.97) [-82.04; 103.27] |
| A4-Smear&LAM vs A3- Smear&Xray | +14 | 932.36 (1044.36) [-1079.67; 3040.75] | 10.24 (185.66) (-376.58; 330.06) | 7.78 (652.37) [-79.70; 87.95] |
| A5- LAM&smear&Xray vs. A3- Smear&Xray | +25 | 2733.23 (932.01) [1031.98; 4655.32] | 344.34 (103.09) (98.91; 511.07) | 8.63 (83.34) [-2.88; 21.41] |
| A4-Smear&LAM vs A6- Xpert_sputum | +12 | 2304.02 (1495.18) [-627.40; 5320.22] | 58.91 (224.51) (-393.45; 479.22) | 5.60 (656.04) [-138.95; 144.29] |
| A7- Xpert_sputum&Xpert_urine vs. A6- Xpert_sputum | +11 | 2460.01 (1029.11) [-449.47; 695.97] | 24.53 (188.64) (-342.98; 393.73) | 2.64 (749.79) [-226.23; 208.12] |
| A8-LAM&Xpert_sputum vs A6- Xpert_sputum | +21 | 2234.92 (1085.30) [196.13; 4453.12] | 91.44 (246.42) (-411.40; 551.03) | 24.02 (982.73) [-114.03; 123.40] |
| A8-LAM&Xpert_sputum vs A7- Xpert_sputum&Xpert_urine | +10 | -225.09 (1377.01) [-3067.17; 2404.47] | 66.90 (248.52) [-435.93; 534.72) | - 12.56 (905.64) [-58.28; 56.91] |
| A7- Xpert_sputum&Xpert_urine vs A9- Xpert_sputum&X-ray | +7 | 787.43 (1078.75) [-1293.07; 2942.31] | -147.16 (220.59) (-609.71; 250.60) | 68.76 (6743.72) [-85.76; 85.89] |
| A10- LAM&Xpert_sputum&Xray vs A9- Xpert_sputum&X-ray | +17 | 2697.17 (933.73) [-972.34; 4609.89] | 219.70 (128.91) (-66.34; 456.55) | 35.52 (1452.88) [-37.85; 68.64] |

†CIs estimated using 10,000 non-parametric bootstrapping replicates; ICER: Incremental Cost-Effectiveness Ratio

relatively high number of TB cases and they were the most cost-effective with a cost/DALY of €4.6 and €4.7 respectively.

Incremental costs and DALYs and the incremental cost effectiveness ratio are shown in Table 4. The algorithms with the highest increase in the cost-effectiveness ratio were replacing smear with LF-LAM and replacing GeneXpert in urine with LF-LAM. Replacing smear with LF-LAM test (A2-LAM versus A1-smear) allowed to detect 23 additional patients and reduced the cost by €87 for an incremental efficiency of 103 DALYs. Replacing GeneXpert in urine with LF-LAM test (A8-LAM&Xpert versus A7-Xpert&Xpert_urine) detected 10 additional patients and reduced the cost by €225 for an incremental efficiency of 70 DALYs.

On the other hand, adding LF-LAM to smear microscopy alone (A4-smear&LAM versus A1-smear) or to smear microscopy and Xray (A5-LAM&smear&Xray versus A3-smear&Xray)

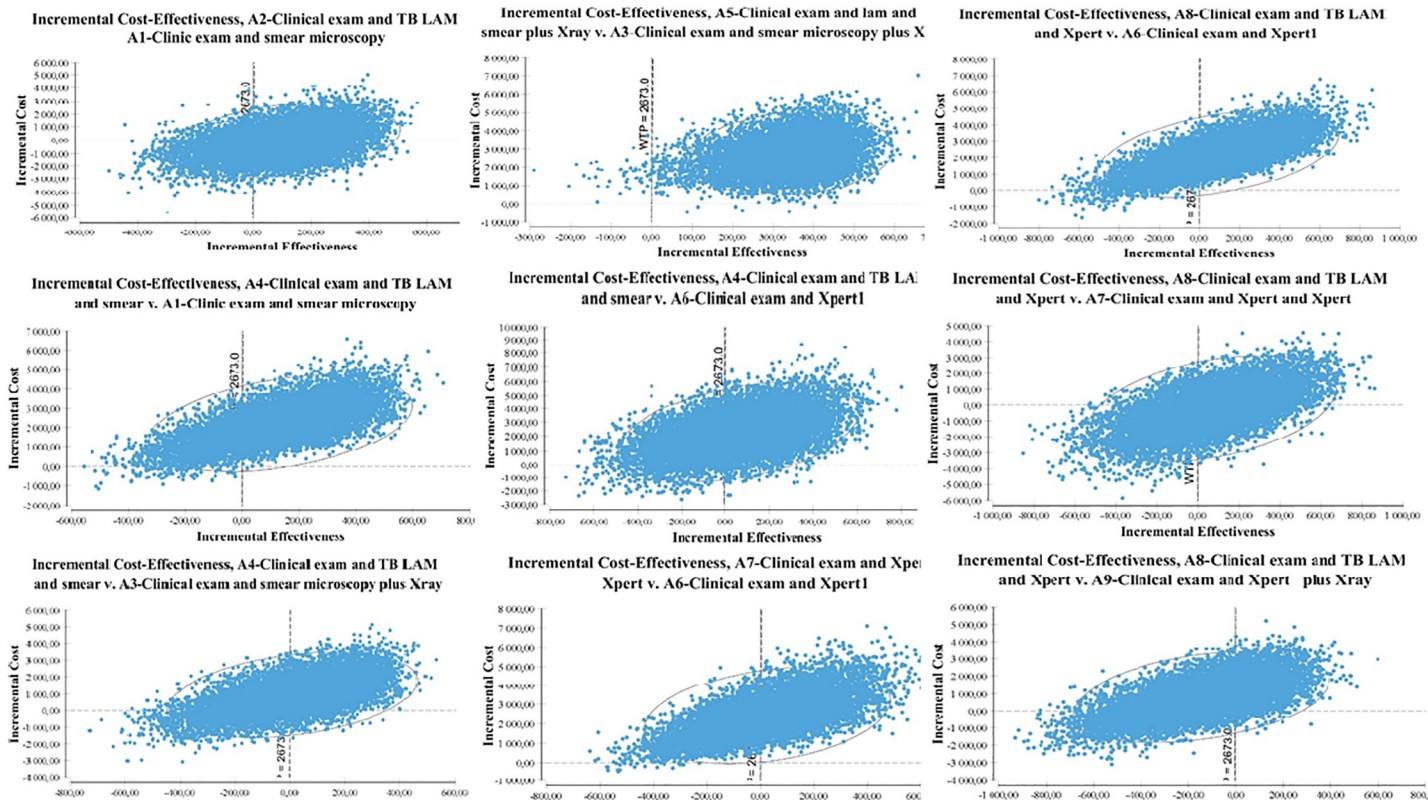

**Fig 2. Cost-effectiveness planes.** This graph represent the differences in costs and DALYs between algorithms. DALYs are plot on the x axis and costs on the y axis. Results in costs and differences in DALYs averted observed in the 10, 000 bootstrap replicates.

detected respectively 31 and 25 additional patients for an incremental cost of €2,219 and €2,733. Their incremental efficiency was estimated respectively at 134 and 344 DALYs.

Fig 2 show the uncertainty assessment from the bootstrap procedure plotted on a cost-effectiveness plane for each algorithm comparison. This figure plot the differences in costs and differences in DALYs averted observed in the 10,000 bootstrap replicates. ICER's cost-effectiveness plane provides a visual representation of the comparison of the two algorithms. The cost-effectiveness plane is thus divided into four quadrants, a north-west-south-east axis where the new strategy is either dominated (it is more expensive and less efficient) or dominant (it is less expensive and more efficient), and a north-east-south-west axis where the decision-maker has to arbitrate between a health gain and a higher cost, or possibly accept a worse result for a lower expense according to the country willingness to pay [38]. ICERs give us the cost/gain of an additional efficiency unit. Introducing LF-LAM test gave us an approximately equivalent ICER when performed in addition to sputum microscopy (A4-smear &LAM versus A3-smear&Xray), performed in replacement of sputum microscopy (A2-LAM versus A1-smear) or done in parallel with sputum microscopy exam and Chest X-ray (A5-LAM&smear&Xray versus A3-smear&Xray), with respectively, €8/DALYs averted; €7/DALYs averted and €9/DALYs averted. The ICER reached €36/DALYs averted when LF-LAM test was performed instead of the GeneXpert test (A10-LAM&Xpert versus A9-Xpert&X-ray) and fell at €3/DALYs averted when an additional GeneXpert urine test was performed in addition to an GeneXpert sputum test (A7-Xpert&Xpert_urine versus A6-Xpert7 versus A6).

Although the cost-effectiveness is done from the DALY, we chose to complement this with calculation of cost/TB case detected in order to provide information about the resource

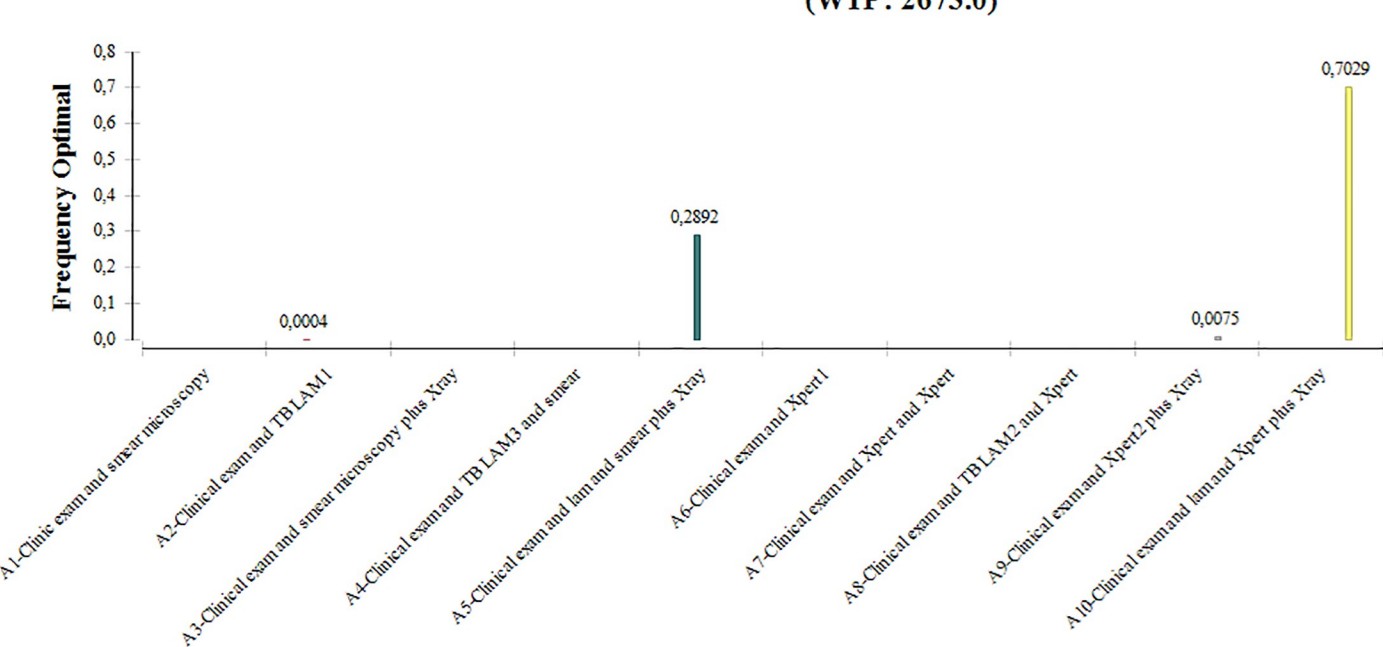

**Fig 3. Probability of cost-effectiveness of each algorithm according to the society's Willingness To Pay (€2,673).**

requirements for implementing the chosen algorithm. The cost/TB case detected ranged from €47 (A2-LAM algorithm) to €70 (A5- LAM&smear+Xray).

The Monte Carlo percent at willingness to pay (€2,673) is shown in Fig 3 where all algorithms were compared together. It represents the probability that one strategy will be efficient compared to all others. This global comparison could orient policy makers on the most efficient algorithm if the set of TB diagnostic tools is available in their country. For a willingness to pay of €2, 673, the acceptability percentage showed that there is 70% of chance that algorithm A10-LAM&Xpert+/-Xray was the most efficient. This percentage dropped to 29% for A5-LAM&smear+/-Xray algorithm. This rate reached 0% for A1-clinic&smear; A3-clinic&smear+/-Xray; A4-smear&LAM; A6-Xpert; A7-Xpert&Xpert_urine and A8-LAM&Xpert algorithms.

The results of the sensitivity analyses do not change our results.

## Discussion

The objective of this study was to rationally and comprehensively evaluate the cost of introduction of LF-LAM in TB diagnostic algorithms for severely ill or immunosuppressed HIV-positive patients. The comparison of the algorithms two by two leads to four main conclusions. Firstly, replacing smear microscopy with LF-LAM test is highly cost-effective. There are 23 additional patients detected for lower cost (€-87) and 103 effective incremental DALYs averted. This result is interesting and encouraging in the prospect of improving access to care in rural and remote areas that do not have access to even basic diagnostic devices, and no trained human resources. Scarcity of water and unreliable electricity supply are additional challenges in setting up smear microscopy. In order to circumvent these problems, LF-LAM

could be an alternative to sputum microscopy for HIV-positive patients as it requires minimal skills and no laboratory equipment. More, LF-LAM is affordable, easy to transport, to use and produces results quickly. Secondly, in the same way, is the possibility to optimize the smear exam by adding the LF-LAM test, where 31 additional patients were detected for an ICER of €22/DALYs averted. In addition, it is preferable to favor the association of smear microscopy with LF-LAM rather than Chest-Xray as more patients were detected even though the ICER was less (€8/DALYs averted). Chest Xray requires expertise for interpreting with high inter-observer variability [39]. However, when this examination exists in addition to the smear, the addition of LF-LAM remains highly cost-effective with 25 additional patients detected and an ICER at €9/DALYs averted. Thirdly, it is more cost-effective to perform smear and LF-LAM tests than GeneXpert test alone and when GeneXpert is available, it is more cost-effective to perform GeneXpert in sputum and LF-LAM test rather than GeneXpert in sputum and GeneXpert in urine. The incremental cost is higher (€2,460 versus €2,234), but the incremental effectiveness is more important (91 DALYs averted versus 25 DALYs averted). The economic evaluation demonstrates that the incorporation of LF-LAM diagnostic tests in the set of TB diagnostic tool is highly cost-effective and complementary to clinical and radiological exams and GeneXpert. LF-LAM can be a real alternative for HIV-positive patients in areas with limited laboratory capacity. The cost-effectiveness analysis showed that the complementarity of all test was highly cost effective given the threshold of €2,673, the three times Kenya/capita annual gross income [27].

We also conducted an additional analysis where the ten algorithms are compared all together (Appendix Figure) in order to know which diagnostic algorithm should be preferred if the states had the set of available tests. We found that the A10-LAM & Xpert+/-Xray algorithm was the most cost-effective algorithm. Indeed, the 10,000 replicates bootstrap results shown that the A10-LAM&Xpert+/-Xray algorithm would be the preferred strategy given a willingness to pay of €2, 673. This algorithm had a 70% probability to be the most efficient. Despite the higher cost of this algorithm, this analysis shows that implementing an algorithm which includes all TB diagnostics tools available is highly cost-effective. This second analysis is interesting for countries with sufficient resources to have all available tests, but also having a sufficient enabling environment to allow the proper use of these tests (human resources, transportation etc.). Our main analysis will apply to countries with limited resources to provide all the tests such as where only sputum microscopy is implemented.

In a context of limited resources and fixed budget, a threshold serves as a reference for decision-makers. This normative threshold of three/capita gross national income implemented by Garber and Phelps [38] is used insofar as it is consistent with accepted practice for economic evaluation. The use of threshold values in decision-making has raised the question of possible uncontrolled growth in health expenditure. However, there are considered relevant as they represent the society's willingness to pay [40].

This is in line with other cost-effectiveness studies of TB diagnostics incorporating the use of LF-LAM test in addition to GeneXpert assay conducted in Uganda [41] and in Uganda and South Africa [42]. In the Uganda study, the combination of GeneXpert with LF-LAM was considered highly cost-effective (ICER $57/DALY-averted) as compared to an algorithm of GeneXpert testing alone. In a study conducted in Uganda and South Africa, Sun et al. [42] found a cost-effectiveness of $353/DALY averted in South Africa and $86/DALY averted in Uganda. Sensitivity analysis confirmed the aforementioned results. In Malawi and South Africa, Reddy et al. [43] found a cost-effectiveness of $450/YLS in Malawi and $840/YLS in South Africa. We calculated of cost/TB case detected to provide an option for countries to identify resource requirements for implementing the chosen algorithm. It gives a basis for budgetary impact evaluation.

This study has some limitations. The first one is related to the study design. The diagnostic algorithms were evaluated from the same patient population in a prospective cohort and were not randomly allocated to different patient groups. Therefore, algorithms were not fully independent from each other, which may have affected their effectiveness outcomes. A second limitation is using culture or GeneXpert confirmed TB as reference for the assessment of the algorithms' performance. This allowed to use a strong reference but excluded from the study population patients without culture or GeneXpert results. Some of them were diagnosed only through LF-LAM. The cost-effectiveness of using LF-LAM could have been higher if patients with symptoms and no culture or GeneXpert results had been included.

## Conclusion

Using urine LF-LAM in addition or as replacement of other diagnostic TB tools is highly cost-effective for severely ill or immunosuppressed HIV-positive patients. A budget impact analysis is needed to guide policy makers.

## Supporting information

**S1 Table. Study database.** For all variables, 1 = « Yes » , 0 = « No »; Sex: 1 = male, 2 = Female; Ageonsetdisease = « age of onset of disease »; Yearbirth = « year of birth »; StandardresidualLE = « Standard residual life expectancy »; age of death HIV/TB = « expected age of death due to the HIV/TB co-infection »; BMI = « Body Mass Index »; lamfinal = « LAM test result »; goldstandall = « TB confirmed »; lamsmear = « algorithm Lam&Smear »; mortality2m = « Death at 2 month »; Cliniclam = « algorithm clinic&lam »; Clinicsmear = « algorithm clinic&smear »; Clinicsmearxray = « algorithm clinic&smear&xray »; Cliniclamsmear = « algorithm clinic&lam&smear »; Clinicxray = « algorithm clinic&xray »; Clinicxraylam = « algorithm clnic&xray&lam »; Clinicsmearxraylam = « algorithm clinic&smear&ray&lam »; clinicxpert1 = « algorithm clinic&xpert sputum »; clinicxpertlam1 = « algorithm clinic&xpert sputum&lam »; clinicxpertxray1 = « algorithm clnic&xray&xpert sputum »; clinicxpertxraylam1 = « algorithm clinic&xray&xpert sputum&lam »; YLD = « Years Lost due to Disability »; YLL = « Years of Life Lost due to premature mortality »; YLD treated clinicLAM = « Nmber of years lost due to screening for algorithm clinic&lam »; YLD treated Clinicsmear = « Number of years lost due to screening for algorithm clinic&smear »; YLD treated Clinicsmearxray = « Number of years lost due to screening for algorithm clinic&smear&xray »; YLD treated Cliniclamsmear = « Number of years lost due to screening for algorithm clinic&lam&smear »; YLD treated Clinicxray = « Number of years lost due to screening for algorithm clinic&xray »; YLD treated Clinicxraylam = « Number of years lost due to screening for algorithm clnic&xray&lam »; YLD treated Clinicsmearxraylam = « Number of years lost due to screening for algorithm clinic&smear&ray&lam »; YLD treated clinicxpert1 = « Number of years lost due to screening for algorithm clinic&xpert sputum »; YLD treated clinicxpertlam1 = « Number of years lost due to screening for algorithm clinic&xpert sputum&lam »; YLD treated clinicxpertxray1 = « Number of years lost due to screening for algorithm clnic&xray&xpert sputum »; YLD treated clinicxpertxraylam1 = « Number of years lost due to screening for algorithm clinic&xray&xpert sputum&lam »; DALYs = « Disability-Adjusted Life Year »; DALYavertedclinicLAM = « DALYs averted with algorithm clinic&lam »; DALYavertedClinicsmear = « DALYs averted with algorithm clinic&smear »; DALYavertedClinicsmearxray = « DALYs averted with algorithm clinic&smear&xray »; DALYavertedCliniclamsmear = « DALYs averted with algorithm clinic&lam&smear »; DALYavertedClinicxray = « DALYs averted with algorithm clinic&xray »; DALYavertedClinicxraylam = « DALYs averted with algorithm clnic&xray&lam »; DALYavertedClinicsmearxraylam = « DALYs averted with algorithm clinic&smear&ray&lam »; DALYavertedclinicxpert1 = « DALYs averted with

algorithm clinic&xpert sputum »; DALYavertedclinicxpertlam1 = « DALYs averted with algorithm clinic&xpert sputum&lam »; DALYavertedclinicxpertxray1 = « DALYs averted with algorithm clnic&xray&xpert sputum »; DALYavertedclinicxpertxraylam1 = « DALYs averted with algorithm clinic&xray&xpert sputum&lam ».
(XLSX)

## Acknowledgments

We are grateful to the patients who participated in this study, and to the *Médecins Sans Frontières* and Ministry of Health staff teams who provided care to the patients and contributed to this study. We would like to thank the Kenyan Ministry of Health and National TB Control Program for their support.

## Author Contributions

**Conceptualization:** Nadia Yakhelef, Martine Audibert, Helena Huerga.

**Data curation:** Nadia Yakhelef.

**Formal analysis:** Nadia Yakhelef.

**Funding acquisition:** Nadia Yakhelef, Helena Huerga.

**Investigation:** Nadia Yakhelef.

**Methodology:** Nadia Yakhelef, Martine Audibert.

**Project administration:** Helena Huerga.

**Resources:** Helena Huerga.

**Software:** Helena Huerga.

**Supervision:** Martine Audibert, Helena Huerga.

**Validation:** Nadia Yakhelef, Martine Audibert, Helena Huerga.

**Visualization:** Nadia Yakhelef, Martine Audibert, Helena Huerga.

**Writing – original draft:** Nadia Yakhelef, Helena Huerga.

**Writing – review & editing:** Nadia Yakhelef, Martine Audibert, Gabriella Ferlazzo, Joseph Sitienei, Steve Wanjala, Francis Varaine, Maryline Bonnet, Helena Huerga.

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
