## [Decision Letter · Decision Letter 0]

1 Oct 2019

PONE-D-19-20635

Cost-effectiveness of diagnostic algorithms including lateral-flow urine lipoarabinomannan for HIV-positive patients with symptoms of tuberculosis

PLOS ONE

Dear Dr. Nadia,

Thank you for submitting your manuscript to PLOS ONE. After careful consideration, we feel that it has merit but does not fully meet PLOS ONE’s publication criteria as it currently stands. Therefore, we invite you to submit a revised version of the manuscript that addresses the points raised during the review process.

We would appreciate receiving your revised manuscript. To enhance the reproducibility of your results, we recommend that if applicable you deposit your laboratory protocols in protocols.io, where a protocol can be assigned its own identifier (DOI) such that it can be cited independently in the future. For instructions see: http://journals.plos.org/plosone/s/submission-guidelines#loc-laboratory-protocols

We look forward to receiving your revised manuscript.

Kind regards,

Frederick Quinn

Academic Editor

PLOS ONE

**Journal Requirements:**

2. We noticed you have some minor occurrence(s) of overlapping text with the following previous publication(s), which needs to be addressed:

https://doi.org/10.5588/ijtld.13.0630

https://doi.org/10.1371/journal.pone.0170976

https://doi.org/10.1371/journal.pmed.1002792

https://dx.doi.org/10.1371%2Fjournal.pone.0117009

https://halshs.archives-ouvertes.fr/halshs-01241824/file/2015.34.pdf

In your revision ensure you cite all your sources (including your own works), and quote or rephrase any duplicated text outside the Methods section. Further consideration is dependent on these concerns being addressed."

3. We note that you have numbered 5 affiliations in your title page, but only assigned 1,2,3, & 4 to authors.  Please assign all the affiliations listed to an author, or remove no.5 if it is not needed.

**Comments to the Author**

1. Is the manuscript technically sound, and do the data support the conclusions?

Reviewer #1: Partly

Reviewer #2: Yes

2. Has the statistical analysis been performed appropriately and rigorously? 

Reviewer #1: Yes

Reviewer #2: Yes

3. Have the authors made all data underlying the findings in their manuscript fully available?

Reviewer #1: Yes

Reviewer #2: Yes

4. Is the manuscript presented in an intelligible fashion and written in standard English?

Reviewer #1: Yes

Reviewer #2: Yes

5. Review Comments to the Author

Reviewer #1: The manuscript entitled “Cost-effectiveness of diagnostic including lateral-flow urine lipoarabinomannan for HIV-positive patients with symptoms of tuberculosis” investigates the cost-effectiveness of including lateral-flow urine lipoarabinomannan (LF-LAM) in TB diagnostic algorithm for severely ill or immunosuppressed HIV-positive patients with symptoms of TB. Ten diagnostic algorithms have been elaborated and compared, and make a conclusion that including LF-LAM in TB diagnostic algorithms is cost-effective for severely ill or immunosuppressed HIV-positive patients.

Series of data have been done by authors to confirm the conclusion. However, there are some drawbacks making the paper to be reconsidered, as detailed below, before it can become suitable for publication in “PLOS ONE”.

Specific comments to the paper:

1. HIV patients often have difficulties to produce sputum, thus only the GeneXpert, LF-LAM seem to be fast and effective ways to detect TB. What the purpose of the research? What the novelty of the research?

2. All the figures in the manuscript were not clear, need to revise.

Reviewer #2: Manuscript PONE-D-19-20635: Cost-effectiveness of diagnostic algorithms including lateral-flow urine lipoarabinomannan for HIV-positive patients with symptoms of tuberculosis

Key Results:

In the manuscript, Yakhelef et al. attempt to determine the cost effectiveness of the implementation of urine LF-LAM test for the detection of Tuberculosis (TB) positive patients whom are severely immunocompromised or are HIV positive using previous data from a study conducted at the Homa Bay County Hospital in Kenya. The authors used ten different algorithms, each containing a different combination of TB treatments starting from a smear alone up to combinations containing a mixture of LF-LAM, Gene Xpert from sputum, Gene Xpert from urine and X-ray detection. Their findings were reported in cost/patient/year in Euros as well as disability adjusted life year (DALY). From their analysis, the authors found that the greatest cost for treatment was anti-tuberculosis drugs (€41.42) and the cheapest costs was the chest X-ray (€1.12). When looking at the total cost/algorithm, the most expensive was A10 consisting of LF-LAM, Xpert from sputum and chest X-ray (€9758), while the least expensive was A2 consisting of LF-LAM alone (€5612). The mean DALY scores were found to be 1118 at its lowest for A1 and 1651 at its highest for A10. Using a cost effectiveness ratio, ICER, the authors determined that the two algorithms with the highest ratios were A2, detecting an additional 23 patients and decreasing costs by €87 as compared to A1, and A8, detecting an additional 10 patients and decreasing costs by €225 as compared to A7. Both of these findings indicate that adding LF-LAM to TB testing may decrease costs while increasing detection. Overall, the authors came to several conclusions: First, that adding LF-LAM testing to current methods is cost effective. Additionally, using LF-LAM in place of chest X-rays has the ability to detect more TB positive patients. Next, replacing Xpert in urine with LF-LAM is more cost effective and lastly, when comparing all of the algorithms to each other, A10 emerges as the most cost effective and most efficient overall if the country had the funds to implement it.

Validity:

After reading this manuscript, I believe that the concepts and findings of this study are valid and novel and may help provide further insight into the reduction of cost of TB diagnosis in resource deprived areas of the world.

Originality:

The results of this manuscript are original and significant. Further studies are needed to incorporate these findings into policy and influence policy makers, however, the authors address this issue in the paper.

Data and Methodology:

The data and methodology of this manuscript are valid and presented in a clear and concise manner. All data is presented clearly and are easy to interpret.

Appropriate use of Statistics:

The statistics in this manuscript are presented properly with a description of methods used, the proper numbers and significance values.

Conclusions:

Based on the experimental design and results presented, the conclusions present as valid and reliable.

Suggested Improvements:

Minor revisions:

- Introduction: This paper focuses on the LF-LAM assay, however, the assay is not described anywhere in the manuscript. Please consider adding a brief overview of the assay to familiarize the reader with what it does.

- Line 110: The title of your figure should give the reader a brief description of what they are looking at. Please consider revising this title as well as those in other figures and tables in the manuscript as many are very sparse in description.

- Cost estimation: Line 129: The acronym IPSOR is used without definition. Please define.

- Cost estimation: Line 163: The acronym CXR is used without definition. Please define.

- Cost estimation: Line 155: “pipette” should be plural in this case, should be “pipettes”

- Cost estimation: Line 164: Here and in several places in the manuscript you indicate that a 4% discount was applied, however, I could not find the reasoning for this discount. Please consider adding an explanation to the manuscript.

- Line 173: “Tables 1” does not need to be plural in this instance, should be “Table 1”

- Study population: Line 199: Are these all of the statistics that are available for the patients? Consider adding additional statistics if possible.

- Study population: Line 200: “patients had a confirmed TB” should not have an “a”. Should read “patients had confirmed TB”

- Line 218: “Tables 3” does not need to be plural. Should read “Table 3”

- Line 249: “Figures 2” does not need to be plural. Should read “Figure 2”

- Line 256: The acronym WTP is used without definition. Please define.

- Lines 274-279: This paragraph is very confusing the way that it is written, please consider revising.

- Line 326: The acronym GNI is used without definition. Please define.

References:

References are cited properly and addressed properly where needed.

Clarity and Context:

The abstract, introduction and conclusion contain the proper information that will engage the reader and provide insight to the paper as a whole.

Scope of Expertise:

None of the material presented were out of the scope of my expertise. All were logical and easily understood.

6. PLOS authors have the option to publish the peer review history of their article (what does this mean?). If published, this will include your full peer review and any attached files.

Reviewer #1: No

Reviewer #2: No

---

## [Author Response · Author response to Decision Letter 0]

23 Nov 2019

Response to Reviewers

PONE-D-19-20635

Cost-effectiveness of diagnostic algorithms including lateral-flow urine lipoarabinomannan for HIV-positive patients with symptoms of tuberculosis

Journal Requirements:

Please ensure that your manuscript meets PLOS ONE's style requirements, including those for file naming. The PLOS ONE style templates can be found at: 

Réponse : we modified according to the requirements of PLOS ONE : 

- The affiliation of the authors and we wrote a legend about the work of each author

- We modified the presentation of figures and tables titles and legends

2. We noticed you have some minor occurrence(s) of overlapping text with the following previous publication(s), which needs to be addressed: :

https://doi.org/10.5588/ijtld.13.0630

https://doi.org/10.1371/journal.pone.0170976

https://doi.org/10.1371/journal.pmed.1002792

https://dx.doi.org/10.1371%2Fjournal.pone.0117009

https://halshs.archives-ouvertes.fr/halshs-01241824/file/2015.34.pdf

In your revision ensure you cite all your sources (including your own works), and quote or rephrase any duplicated text outside the Methods section. Further consideration is dependent on these concerns being addressed. 

Response: Thank you, we modify as follows: 

- https://doi.org/10.5588/ijtld.13.0630 � This reference has been added [19] (lines 148, 157, 174 of the manuscript) and we modify the passages where there were certain occurrences (lines 148 to 154 and 179 to 182 of the manuscript)

- https://doi.org/10.1371/journal.pone.0170976 �This reference [15] has been added lines 78 and 110 of the manuscript 

- ttps://halshs.archives-ouvertes.fr/halshs-01241824/file/2015.34.pdf � we modify the sections where there were certain occurrences (lines 355 to 357 of the manuscript)

- https://dx.doi.org/10.1371%2Fjournal.pone.0117009 �This reference [30] has been added lines 206

3. We note that you have numbered 5 affiliations in your title page, but only assigned 1,2,3, & 4 to authors. Please assign all the affiliations listed to an author, or remove no.5 if it is not needed 

Response: Thank you, the mistake has been corrected. 

Response: thanks for the comment, it is well noted

We have added a paragraph “supporting information” line 505-507

5. Review Comments to the Author

Reviewer #1: 

1. HIV patients often have difficulties to produce sputum, thus only the GeneXpert, LF-LAM seem to be fast and effective ways to detect TB. What the purpose of the research? What the novelty of the research? 

Response:

Thank you for this question. To enlighten the reader we have added a paragraph on the purpose as well as the novelty of this research at the end of the introduction (lines 85 to 88 of the manuscript) 

2. All the figures in the manuscript were not clear, need to revise. 

Response: Thank you for this remark. We have improved the quality of the figures. In case our article is accepted, we will also discuss with PLOS One production team how to improve further the quality of the figures. 

Reviewer #2: Manuscript PONE-D-19-20635: Cost-effectiveness of diagnostic algorithms including lateral-flow urine lipoarabinomannan for HIV-positive patients with symptoms of tuberculosis

- Introduction: This paper focuses on the LF-LAM assay, however, the assay is not described anywhere in the manuscript. Please consider adding a brief overview of the assay to familiarize the reader with what it does.

Response: Thank you for this proposition, we have added a brief overview of the assay (lines 62 to 71 of the manuscript)

- Line 110: The title of your figure should give the reader a brief description of what they are looking at. Please consider revising this title as well as those in other figures and tables in the manuscript as many are very sparse in description. 

Response: Thank you for this remark, we agree. We have changed Figure titles as follows:

- Fig 1. Diagnostic algorithms using microscopy or GeneXpert alone or in combination with chest X-ray versus LF-LAM alone or in addition to other TB diagnostic tools (lines 125-126 of the manuscript)

- Fig 2. Cost-effectiveness planes (This graph represent the differences in costs and DALYs between algorithms. DALYs are plot on the x axis and costs on the y axis. Results in costs and differences in DALYs averted observed in the 10, 000 bootstrap replicates) lines 296 to 298 of the manuscript)

- Fig 3. Probability of cost-effectiveness of each algorithm according to the society’s Willingness To Pay (WTP) (€2,673) (lines 308-309 of the manuscript)

- Cost estimation: Line 129: The acronym IPSOR is used without definition. Please define. 

Response : Thank you for this remark, the acronym has been defined (lines 145-146 of the manuscript)

- Cost estimation: Line 163: The acronym CXR is used without definition. Please define.

Response : Thank you for this remark, we have changed CXR by Chest X-ray. (line 180 of the manuscript)

- Cost estimation: Line 155: “pipette” should be plural in this case, should be “pipettes”

Response :Thank you for this remark, we have added an « s » to the word pipettes (line 172 of the manuscript)

- Cost estimation: Line 164: Here and in several places in the manuscript you indicate that a 4% discount was applied, however, I could not find the reasoning for this discount. Please consider adding an explanation to the manuscript. 

Response :. Thank you for this comment. We have added an explanation and justified the rate discount chosen which is a recommendation of the French Health Authority (lines 181-182 and 189 of the manuscript)

- Line 173: “Tables 1” does not need to be plural in this instance, should be “Table 1”

Response: Thank you , we have deleted the « s » (line 192 of the manuscript)

- Study population: Line 199: Are these all of the statistics that are available for the patients? Consider adding additional statistics if possible.

Response: Thank you. We have included some additional description of the population as suggested (lines 218 to 225 of the manuscript)

- Study population: Line 200: “patients had a confirmed TB” should not have an “a”. Should read “patients had confirmed TB”

Response : Thank you , we have deleted the « a » (line 226 of the manuscript)

- Line 218: “Tables 3” does not need to be plural. Should read “Table 3”

Response : Thank you , we have deleted the « s » (line 244 of the manuscript)

- Line 249: “Figures 2” does not need to be plural. Should read “Figure 2”

Response : Thank you, we have removed the « s » (line 275 of the manuscript)

- Line 256: The acronym WTP is used without definition. Please define.

Response : Thank you, we have removed the acronym all over the manuscript and used the full word and, we propose to add a definition at this level (lines 202-203 of the manuscript).

- Lines 274-279: This paragraph is very confusing the way that it is written, please consider revising.

Response: Thank you for this comment. We have modified the paragraph as suggested (line 299 to 306 of the manuscript) 

- Line 326: The acronym GNI is used without definition. Please define.

Response: Thank you for this comment. Thank you, we have removed the acronym all over the manuscript and used the full word.

---

## [Decision Letter · Decision Letter 1]

3 Dec 2019

PONE-D-19-20635R1

Cost-effectiveness of diagnostic algorithms including lateral-flow urine lipoarabinomannan for HIV-positive patients with symptoms of tuberculosis

PLOS ONE

Dear Dr. Nadia,

Thank you for submitting your manuscript to PLOS ONE. After careful consideration, we feel that it has merit but does not fully meet PLOS ONE’s publication criteria as it currently stands. Therefore, we invite you to submit a revised version of the manuscript that addresses the points raised during the review process.

We would appreciate receiving your revised manuscript. To enhance the reproducibility of your results, we recommend that if applicable you deposit your laboratory protocols in protocols.io, where a protocol can be assigned its own identifier (DOI) such that it can be cited independently in the future. For instructions see: http://journals.plos.org/plosone/s/submission-guidelines#loc-laboratory-protocols

We look forward to receiving your revised manuscript.

Kind regards,

Frederick Quinn

Academic Editor

PLOS ONE

Reviewers' comments:

Reviewer's Responses to Questions

**Comments to the Author**

1. If the authors have adequately addressed your comments raised in a previous round of review and you feel that this manuscript is now acceptable for publication, you may indicate that here to bypass the “Comments to the Author” section, enter your conflict of interest statement in the “Confidential to Editor” section, and submit your "Accept" recommendation.

Reviewer #1: All comments have been addressed

Reviewer #2: All comments have been addressed

2. Is the manuscript technically sound, and do the data support the conclusions?

Reviewer #1: Yes

Reviewer #2: Yes

3. Has the statistical analysis been performed appropriately and rigorously? 

Reviewer #1: Yes

Reviewer #2: Yes

4. Have the authors made all data underlying the findings in their manuscript fully available?

Reviewer #1: Yes

Reviewer #2: Yes

5. Is the manuscript presented in an intelligible fashion and written in standard English?

Reviewer #1: Yes

Reviewer #2: Yes

6. Review Comments to the Author

Reviewer #1: The revised manuscrip covered all my comments, while there are still exist several minimal mistakes in the revised mauscript.

minimal comments:

1. In line 152, there are two dots, remove one.

2. In the revise manuscript, several references "underline", remove the underline. such as, in line 181, 185, 189, et al.

3. In line 156, reference [17], [19], should be changed into [17, 19].

4. In line 158, in the sentence “number of patients/day", should be changed into "number of patients per day".

5. In line 160, "First" should be changed into "Firstly".

6. In Table 1, in the setences "Age at oneset of disablity a" and "Age of death a", what the word a stand for?

7. In the line 201, the reference [23-24], shoukd be changed into [23, 24].

8. In the line 213, the reference [17-18], shoukd be changed into [17, 18].

9. In the line 222, 100~199 cells/u, should be changed into 100~199 cells/uL.

10. The references in this manuscript should be adjust to fit the PloS One. I saw many differences in the references, such as reference 10, 201-9, should be changed into 201-209. Please doubcheck the references.

Reviewer #2: None. The authors have addressed all of the suggestions that were put forth by me in the initial review. Thank you.

7. PLOS authors have the option to publish the peer review history of their article (what does this mean?). If published, this will include your full peer review and any attached files.

Reviewer #1: No

Reviewer #2: No

---

## [Author Response · Author response to Decision Letter 1]

4 Dec 2019

Response to Reviewers

PONE-D-19-20635R1

Cost-effectiveness of diagnostic algorithms including lateral-flow urine lipoarabinomannan for HIV-positive patients with symptoms of tuberculosis

PLOS ONE

 6. Review Comments to the Author

Reviewer #1: The revised manuscrip covered all my comments, while there are still exist several minimal mistakes in the revised mauscript.

minimal comments:

1. In line 152, there are two dots, remove one.

Response: Thank you , we have deleted the dots (line 152 of the manuscript)

2. In the revise manuscript, several references "underline", remove the underline. such as, in line 181, 185, 189, et al.

Response : Thank you for this remark, we have modified all “underline” references

3. In line 156, reference [17], [19], should be changed into [17, 19].

Response : Thank you for this remark, we have changed reference [17] [19] by [17,19] (line 156 of the manuscript)

4. In line 158, in the sentence “number of patients/day", should be changed into "number of patients per day".

Response : Thank you for this remark, we have changed “number of patients/day” by “number of patients per day” (line 158 of the manuscript)

5. In line 160, "First" should be changed into "Firstly".

Response : Thank you for this remark, we have changed “First” by “Firstly” (line 160 of the manuscript)

6. In Table 1, in the setences "Age at oneset of disablity a" and "Age of death a", what the word a stand for?

Response: Thank you for this careful reading, this is a mistake, we have deleted the “a” (table 1 of the manuscript)

7. In the line 201, the reference [23-24], shoukd be changed into [23, 24].

Response : Thank you for this remark, we have changed reference [23-24] by [23, 24] (line 199 of the manuscript)

8. In the line 213, the reference [17-18], shoukd be changed into [17, 18].

Response : Thank you for this remark, we have changed reference [17-18] by [17, 18] (line 211 of the manuscript)

9. In the line 222, 100~199 cells/u, should be changed into 100~199 cells/uL.

Response : Thank you for this remark, we have changed 100~199 cells/u by 100~199 cells/uL (line 220 of the manuscript)

10. The references in this manuscript should be adjust to fit the PloS One. I saw many differences in the references, such as reference 10, 201-9, should be changed into 201-209. Please doubcheck the references.

Response : Thank you for this remark, we have adjusted the following refeneces : 6, 10, 23, 28, 35, 39, 40, 42 

Reviewer #2: None. The authors have addressed all of the suggestions that were put forth by me in the initial review. Thank you.

7. PLOS authors have the option to publish the peer review history of their article (what does this mean?). If published, this will include your full peer review and any attached files.

Response : Thank you, this is well noted

---

## [Decision Letter · Decision Letter 2]

13 Dec 2019

Cost-effectiveness of diagnostic algorithms including lateral-flow urine lipoarabinomannan for HIV-positive patients with symptoms of tuberculosis

PONE-D-19-20635R2

Dear Dr. Nadia,

We are pleased to inform you that your manuscript has been judged scientifically suitable for publication and will be formally accepted for publication once it complies with all outstanding technical requirements.

With kind regards,

Frederick Quinn

Academic Editor

PLOS ONE

Additional Editor Comments (optional):

Reviewers' comments:

Reviewer's Responses to Questions

**Comments to the Author**

1. If the authors have adequately addressed your comments raised in a previous round of review and you feel that this manuscript is now acceptable for publication, you may indicate that here to bypass the “Comments to the Author” section, enter your conflict of interest statement in the “Confidential to Editor” section, and submit your "Accept" recommendation.

Reviewer #1: All comments have been addressed

Reviewer #2: All comments have been addressed

2. Is the manuscript technically sound, and do the data support the conclusions?

Reviewer #1: Yes

Reviewer #2: Yes

3. Has the statistical analysis been performed appropriately and rigorously? 

Reviewer #1: Yes

Reviewer #2: Yes

4. Have the authors made all data underlying the findings in their manuscript fully available?

Reviewer #1: Yes

Reviewer #2: Yes

5. Is the manuscript presented in an intelligible fashion and written in standard English?

Reviewer #1: Yes

Reviewer #2: Yes

6. Review Comments to the Author

Reviewer #1: The revised manuscript covered all my comments.I have no comments to the authors. Thus, I recommend to accept this manuscript.

Reviewer #2: The authors had addressed my issues and concerns in a previous revision. I have no additional comments.

7. PLOS authors have the option to publish the peer review history of their article (what does this mean?). If published, this will include your full peer review and any attached files.

Reviewer #1: No

Reviewer #2: No

---

## [Editor Report · Acceptance letter]

7 Jan 2020

PONE-D-19-20635R2 

Cost-effectiveness of diagnostic algorithms including lateral-flow urine lipoarabinomannan for HIV-positive patients with symptoms of tuberculosis 

Dear Dr. Yakhelef:

I am pleased to inform you that your manuscript has been deemed suitable for publication in PLOS ONE. Congratulations! Your manuscript is now with our production department. 

With kind regards,

on behalf of

Dr. Frederick Quinn 

Academic Editor

PLOS ONE